# Donor-delivered cell wall hydrolases facilitate nanotube penetration into recipient bacteria

Amit K. Baidya [1], Ilan Rosenshine[1] & Sigal Ben-Yehuda [1✉]

Bacteria can produce membranous nanotubes that mediate contact-dependent exchange of molecules among bacterial cells. However, it is unclear how nanotubes cross the cell wall to emerge from the donor or to penetrate into the recipient cell. Here, we report that *Bacillus subtilis* utilizes cell wall remodeling enzymes, the LytC amidase and its enhancer LytB, for efficient nanotube extrusion and penetration. Nanotube production is reduced in a *lytBC* mutant, and the few nanotubes formed appear deficient in penetrating into target cells. Donor-derived LytB molecules localize along nanotubes and on the surface of nanotube-connected neighbouring cells, primarily at sites of nanotube penetration. Furthermore, LytB from donor *B. subtilis* can activate LytC of recipient bacteria from diverse species, facilitating cell wall hydrolysis to establish nanotube connection. Our data provide a mechanistic view of how intercellular connecting devices can be formed among neighbouring bacteria.

[1] Department of Microbiology and Molecular Genetics, Institute for Medical Research Israel-Canada, The Hebrew University-Hadassah Medical School, The Hebrew University of Jerusalem, POB 12272, 91120 Jerusalem, Israel. ✉email: sigalb@ekmd.huji.ac.il

Bacteria growing in close proximity exhibit networks of contact-dependent interactions, enabling the targeted delivery of DNA, proteins, and small molecules[1,2]. By employing the Gram-positive soil bacterium *Bacillus subtilis* (*Bs*) as a model, we have previously described a type of contact-dependent molecular trade mediated by intercellular membranous bridges, termed nanotubes[3]. Nanotubes were found to serve as conduits for intercellular exchange of metabolites, cytoplasmic proteins and even plasmids, hinting at their potential significance in shaping natural bacterial communities[3–6]. Further, utilizing these intercellular bridges, *Bs* could deliver the WapA tRNase toxin to, and acquire nutrients from, the opponent species *B. megaterium* (*Bm*)[6]. Intriguingly, nanotube-like structures were shown to facilitate trafficking of nutrients among various species, including Gram-positive and Gram-negative bacteria[7,8], and even in a trans-kingdom manner between bacterial pathogens and mammalian host cells[9].

We previously provided evidence that a calcineurin-like phosphodiesterase, YmdB[10,11], is enriched in the nanotube biochemical fraction of *Bs*[5]. In fact, cells lacking *ymdB* were perturbed in nanotube production, a deficiency that coincided with decreased molecular exchange, thereby providing the first genetic link between both processes[5]. More recently, we have demonstrated that five conserved membrane proteins (FlhA, FlhB, FliP, FliQ, and FliR), components of the flagellar export apparatus, designated CORE, serve as a platform for nanotube biogenesis. CORE-dependent nanotube production was shown to be conserved among distinct species, signifying nanotube as a ubiquitous bacterial organelle. In accord, COREs from diverse species could restore nanotube generation and functionality in bacteria lacking endogenous CORE[9,12].

Ultrastructural analysis indicated *Bs* nanotubes to be membranous in nature and to directly emanate from the cell membrane, cross the cell wall (CW) and protrude from the cell surface[5]. The bacterial CW is primarily comprised of peptidoglycan (PG) layers wrapped around the cytoplasmic membrane[13]. In most bacteria, the PG polymer is made of glycan strands cross-linked by peptide side chains which provide strength and rigidity to maintain the cell shape[14,15]. The estimated thickness of the Gram-positive CW is ~30–100 nm, and the effective pore size is ~2 nm (refs. [16–19]). This brings about the questions as of how do the delicate membranous nanotubes emerge from the thick CW shield, and how do they penetrate the protective CW of nearby bacteria. Actually, opening of the PG mesh to insert a new substance into the PG layer is a complex process, mediated by an array of CW hydrolases with tightly regulated activity to avoid the cell's own lysis[20–22]. These hydrolases participate in key cellular processes including cell division and assembly of complex structures, such as secretion systems, flagella, and pili[21,23,24].

Interestingly, the nanotube biochemical fraction of *Bs* was found to contain two major CW hydrolases, LytC (*N*-acetylmuramoyl-L-alanine amidase) and its activator LytB, encoded by the same operon, both shown to be part of the CW associated proteome[5,25–27]. LytC harbors a catalytic domain, which is conserved among Gram-positive and Gram-negative bacteria, that cleaves the amide bond between the *N*-acetyl muramic acid and stem peptide joining the two glycan layers, an activity that is facilitated in the presence of LytB[26]. Notably, LytC was found to impact *Bs* swarming motility, independently of flagella formation[28]; however, the exact role of these CW hydrolases is still elusive.

Here we show that LytB and LytC assist nanotube extrusion from the surface of the producing bacterium. Furthermore, LytB molecules are found to localize over nanotube structures, suggesting that they can travel along nanotubes to reach the surface

of adjacent bacterium of the same or different species. We further provide evidence that interspecies pairs of LytB, from a nanotube producer, and LytC, from a nanotube receiver, can cooperate to promote nanotube penetration to a recipient bacterium. Our results suggest that the repertoire of hydrolases possessed by the interacting bacteria determines the efficiency of interspecies nanotube formation.

## Results

**LytB and LytC impact nanotube extrusion and penetration**. As LytB and LytC hydrolases were found to be associated with the nanotube biochemical fraction[5], we reasoned that they might facilitate the passage of nanotubes through the thick CW material, enabling their reach to the cell exterior. To investigate this possibility, we examined nanotube production in *Bs* mutants lacking *lytB*, *lytC,* or both (Supplementary Fig. 1A, B), grown on a solid surface, by employing Extreme-High Resolution Scanning Electron Microscopy (XHR-SEM). Although intercellular nanotubes were readily detectable in wild-type (wt) cells, their occurrence was significantly reduced in the inspected mutant strains, with the Δ*lytBC* mutant showing the most severe phenotype (Fig. 1a–c; Supplementary Fig. 1C). Consistently, these mutants were deficient in nanotube-associated intercellular molecular trafficking (Supplementary Fig. 2). Interestingly, the few nanotubes that eventually extruded from the Δ*lytBC* mutant cells, appeared as they failed to penetrate into the nearby recipient bacterium, exhibiting unusual "continual" nanotubes, extending over the surface of their neighboring cell (Fig. 1b, d; Supplementary Fig. 1C). Such continual nanotubes were rarely observed in wt cells (Fig. 1a, d; Supplementary Fig. 1C). Similar results were obtained when the amidase active site of LytC was mutated to abolish its catalytic activity in a strain lacking *lytB*, indicating that deficiency in CW degradation is associated with the observed phenotypes (Supplementary Fig. 3). These results hinted that LytB and LytC facilitate extrusion of nanotubes by the producer cell along with their penetration into the receiver bacterium.

**Revealing LytB and LytC localization pattern**. To explore the underlying process of Lyt-mediated nanotube formation, LytB and LytC were tagged with GFP and their localization pattern followed. Both protein fusions were undetectable by fluorescence microscopy (not shown), however, adding antibodies against GFP to live intact cells exposed LytB to be present on the cell surface, arrayed in a foci-like pattern (Fig. 2), while LytC-GFP was still undetectable (Supplementary Fig. 4A). LytB surface localization was promoted upon overexpression of the nanotube-inducing esterase YmdB[5], and dramatically decreased in its absence (Fig. 2). The overall LytB protein levels as well as *lyt* promoter activity were hardly affected by YmdB (Supplementary Fig. 4B, C), implying that only the LytB surface presentation is affected by the esterase. Curiously, LytB-foci were still evident in a Δ*CORE* strain devoid of nanotubes[12] (Supplementary Fig. 4D) with relatively low abundance, indicating that the protein could still reach the cell surface in the absence of nanotubes.

To explore the possibility that LytB surface localization is associated with nanotube generation, we aimed to co-visualize nanotubes and LytB molecules at high resolution. To this end, LytB-GFP expressing cells were subjected to immuno-XHR-SEM, utilizing primary anti-GFP antibodies followed by gold-labeled secondary antibodies. LytB molecules were observed along nanotube structures and over the bacterial surface, mainly close to nanotube sites (Fig. 3a; Supplementary Fig. 5A). These high densities of protein molecules likely correspond to the LytB foci observed on the bacterium surface by immunofluorescence microscopy (Fig. 2a). Similar results were obtained with LytB-

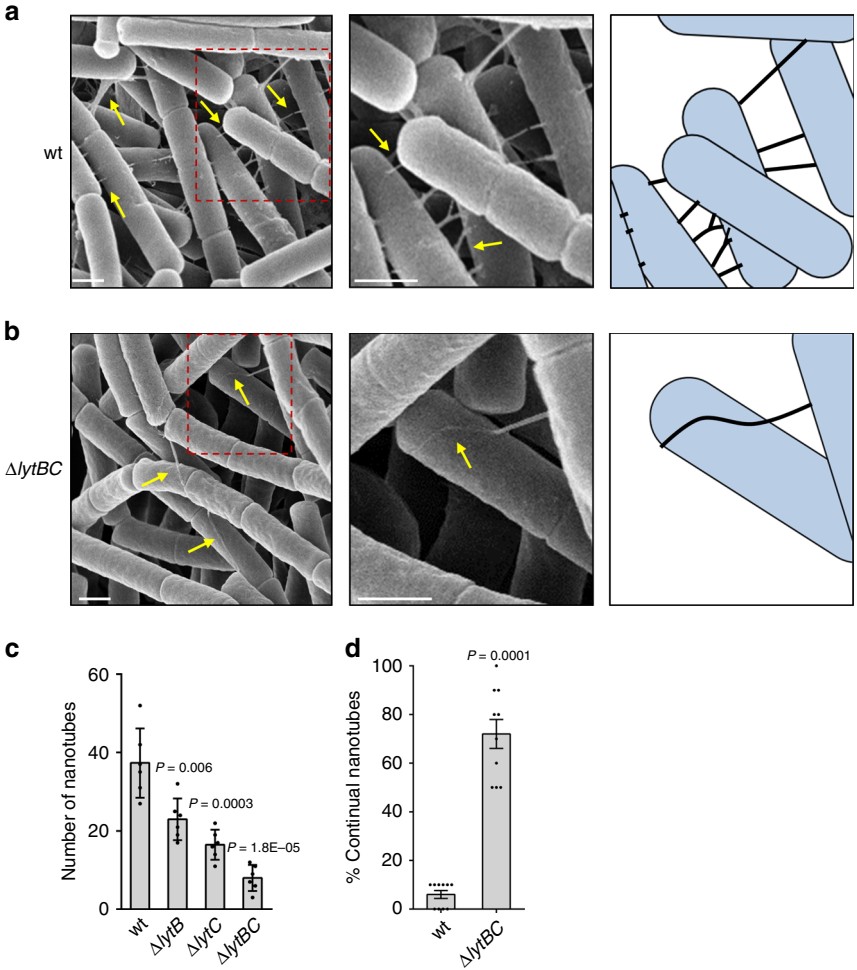

**Fig. 1 LytB and LytC impact nanotube extrusion and penetration. a**, **b** *Bs* wt (GD215: Δ*hag*) (**a**) and Δ*lytBC* (AB64: Δ*lytBC*, Δ*hag*) (**b**) strains were grown to mid logarithmic phase, spotted onto EM grids, incubated on LB agar plates for 4 h at 37 °C, and visualized by XHR-SEM. Shown are typical fields of cells (left panels), enlargements of the boxed regions (middle panels), and schematics that depict the interpretive cell layout and highlight intercellular nanotube patterns (right panels). Arrows indicate intercellular nanotubes. Scale bars represent 500 nm. **c** Quantification of the number of nanotubes displayed per 50 cells. *Bs* wt (GD215 Δ*hag*), Δ*lytB* (AB232: Δ*lytB*, Δ*hag*), Δ*lytC* (AB62: Δ*lytC*, Δ*hag*), and Δ*lytBC* (AB64: Δ*lytBC*, Δ*hag*) strains were subjected to XHR-SEM analysis as described in **a**, **b**. Shown are mean ± SEM and *P* values (unpaired Student's *t* test) of at least three independent experiments ($n_{cells} = 300$). **d** Quantification of the average percentage of continual nanotubes displayed by *Bs* wt (GD215) and Δ*lytBC* (AB64) strains following XHR-SEM analysis as described in **a**, **b**. Shown are mean ± SEM and *P* values (unpaired Student's *t* test) of at least three independent experiments. ($n_{nanotubes} = 150$). Source data are provided as a Source Data file.

HA producing strain stained with anti-HA antibodies (Supplementary Fig. 6A). As controls, gold signal was not obtained from cells lacking GFP or from cells expressing a cytoplasmic ribosomal protein fused to GFP (Supplementary Fig. 6B, C). Furthermore, cells harboring a YFP fusion to an unrelated surface exposed membrane protein yielded a scattered pattern, which did not coincide with nanotubes[12,29] (Supplementary Fig. 6D). Inspecting LytB localization in a *CORE* deficient mutant, revealed LytB molecules to be localized in clusters over the cell surface, consistent with the immunofluorescence results (Supplementary Figs. 4D and 7A). Visualizing LytC revealed fewer molecules in comparison to that of LytB, with the majority being scattered over the cell surface (Supplementary Fig. 7B). Very few LytC molecules were found to localize into nanotube structures (Supplementary Fig. 7B, C), suggesting that, unlike LytB, the LytC enzyme operates at a lower concentration and is not efficiently delivered via nanotubes. Consistently, the signal from cell surface associated LytC, detected by western blot analysis, was significantly weaker than that of LytB (Supplementary Fig. 7D).

**Bs-derived LytB can be delivered to other species via nanotubes.** The localization of LytB molecules to intercellular nanotubes suggests that these molecules may travel via nanotubes to reach the target bacterium and assist in invading the neighboring recipient. To investigate this possibility, we examined whether LytB molecules can be deposited on the surface of a recipient bacterium belonging to a different species. Hence, we co-incubated *Bs*, producing LytB-GFP, with a wt *Bm*, lacking GFP, and carried out Immuno-XHR-SEM analysis to visualize single LytB molecules. Since *Bm* is significantly larger than *Bs*, we could readily distinguish between these two species in the co-culture[6] (Supplementary Fig. 8A). Remarkably, LytB molecules were plentiful on the surface of *Bm* when co-incubated with *Bs*, whereas no gold signal was found on *Bm* surface grown alone (Fig. 3b; Supplementary Figs. 5B and 8B). Furthermore, these LytB molecules were arrayed over the *Bm* surface in vicinity to sites of interspecies connecting nanotubes, but scarce on *Bm* surface regions that were distant from these sites (Fig. 3b; Supplementary Fig. 5B). This phenomenon was emphasized when *Bs*

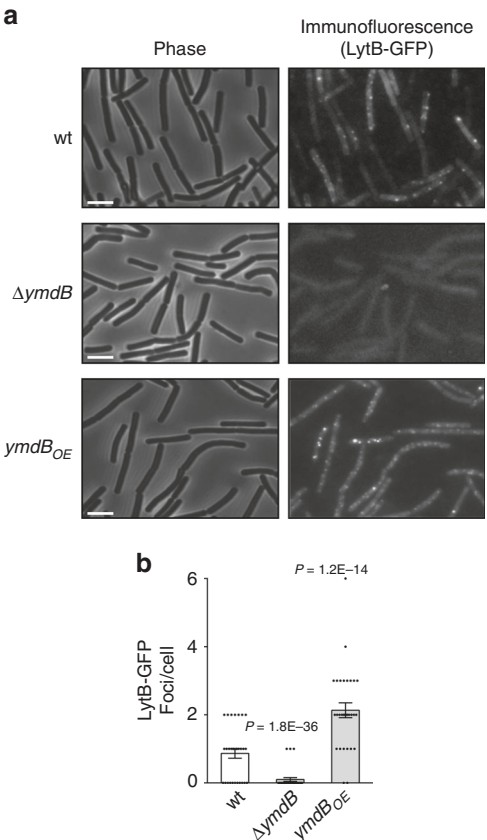

**Fig. 2 LyB localizes to the cell surface in YmdB-dependent manner. a, b** *Bs* wt (AB132: *lytB-gfp*), Δ*ymdB* (AB134: *lytB-gfp*, Δ*ymdB*) and *ymdB*<sub>OE</sub> (AB144: *lytB-gfp*, Δ*ymdB*, *amyE*::P<sub>hyper-spank</sub>-*ymdB*) strains were spotted onto poly L-lysine coated coverslips, treated with anti-GFP primary antibodies and FITC-conjugated secondary antibodies and visualized by fluorescence microscopy. Shown are phase contrast images (left panels) and respective immunofluorescence images (right panels). Scale bars represent 2 μm (**a**). Graph presents quantification of the average number of LytB-GFP foci displayed per cell. Shown are mean ± SEM and *P* values (unpaired Student's *t* test) of at least three independent experiments (n<sub>cells</sub> = 600) (**b**). Source data are provided as a Source Data file.

was co-incubated with the relatively distant coccus bacterium *Staphylococcus aureus* (*Sa*) (Supplementary Fig. 8A). The delivery of LytB molecules was prominent over the surface of *Sa*, particularly in areas proximal to nanotube attachment sites (Fig. 3c; Supplementary Figs. 5C, 8C and 9). Nanotube-like structures were also detected when *Bs* was co-cultured with the morphologically distinct Gram-negative *Vibrio cholerae* (*Vc*) (Supplementary Figs. 8A and 10). Intriguingly, as *Vc* lacks exposed CW, LytB molecules were detected on nanotubes formed between *Bs* and *Vc*, but were yet absent from the *Vc* surface (Fig. 3d; Supplementary Figs. 5D and 8D). These results suggest that LytB molecules are delivered along nanotubes to facilitate nanotube penetration through the CW of Gram-positive bacteria. Such interspecies deposition of molecules was not apparent with LytC, possibly due to the low amount of this protein.

**LytB CW binding domains enable its intra- and inter-species delivery**. Our analysis so far revealed that LytB exhibits three main localization patterns: the cell surface of the producing cell, intercellular nanotubes, and cell surface of the target bacterium. LytB is composed of N-terminus signal peptide (SP) and three

CW binding (CWB) domains[25,27] (Fig. 4a). We therefore sought to delineate which of the protein domains is required for each localization pattern. As a first step, we deleted the LytB SP sequence (LytB-ΔSP). Although the intracellular protein levels were higher in comparison to that of the wt (Fig. 4b), the truncation greatly reduced the placement of the protein from both cell surface and nanotubes, as indicated by immunofluorescence and immuno-XHR-SEM (Fig. 4c, f; Supplementary Fig. 11A, D). Next, we investigated the necessity of LytB CWB domains for its positioning by deleting all three CWB domains (LytB-ΔCWB) (Fig. 4a). Cell surface display of LytB-ΔCWB was largely reduced in comparison to that of wt LytB; nevertheless, localization of the mutant protein to nanotubes was still evident (Fig. 4b, d, f; Supplementary Fig. 11B, D), signifying that the cues for cell surface and nanotube positioning differ. Moreover, co-culturing *Bs* with *Sa* showed that although LytB-ΔCWB molecules were clearly localized with interspecies nanotubes, deposition of the mutant protein on the *Sa* surface was significantly lessened, and the nanotubes appeared to be crawling over the *Sa* surface (Fig. 4e, g; Supplementary Fig. 11C). Consistently, *Bs* cells expressing LytB-ΔCWB mutant form of the protein showed the continual nanotube phenotype in the absence of *lytC* (Supplementary Fig. 12), similar to Δ*lytBC* mutant cells (Fig. 1b). Based on these data, we suggest that LytB CWB domains are dispensable for LytB trafficking along the tube but required for deposition of the protein on the CW of the recipient bacterium to enable efficient nanotube penetration.

**LytB-*Bs* and LytC-*Bs* can act exogenously on CW of foreign species**. The findings that LytB-*Bs* is delivered via nanotubes to the target cell, suggests that it might act on the cell surface to locally facilitate CW degradation from the outside, enabling nanotube penetration to the recipient cell. To explore this possibility, we investigated the capacity of purified LytB and LytC to damage the CW of various bacterial species. Initially, purified LytB-*Bs* and LytC-*Bs* were tested for their activity on isolated *Bs* CW, using Remazol Brilliant Blue (RBB) assay, in which CW degradation activity is reported by dye release[30]. Consistent with previous findings[26,31], only a minute degradation could be detected by LytB-*Bs*, nevertheless the addition of LytC-*Bs* boosted this activity to a level higher than each of the proteins added individually (Fig. 5a). In accordance, addition of the purified proteins to the isolated *Bs* CW led to the formation of visible punctures over the sacculi (Fig. 5b). Furthermore, XHR-SEM analysis showed that the addition of both purified proteins to living cells resulted in severe CW perturbation (Supplementary Fig. 13A). To obtain a quantitative measurement of LytB-*Bs* and LytC-*Bs* activity on intact cells, we incubated living bacteria with the purified enzymes, and assayed the ability of the non-permeable red fluorescent stain propidium iodide (PI) to penetrate the cells. This is based on the observation that CW damage, caused by lysozyme treatment, gradually increases permeability to PI (Supplementary Fig. 13B). A notable correlation between the PI staining and the LytB-*Bs* and LytC-*Bs* activity, as detected by RBB and XHR-SEM, was evident (Fig. 5c, d). Next, we tested the impact of exogenous LytB-*Bs* and LytC-*Bs* on the foreign *Bm* and *Sa* CWs. The activity of purified LytB-*Bs* and LytC-*Bs*, added separately or concurrently, on the CW isolated from *Bm* and *Sa* was almost indistinguishable from their activity on *Bs* CW, as indicated by the RBB assay (Fig. 5a). PI staining, for monitoring CW perturbation revealed strong damage to both bacterial strains following the addition of both LytB-*Bs* and LytC-*Bs* (Fig. 5c, d). Notably, although the activity of sole LytB-*Bs* on purified CW was negligible (Fig. 5a), treating intact *Bs* or *Bm* with this protein showed significant CW impairment, which was not detected for

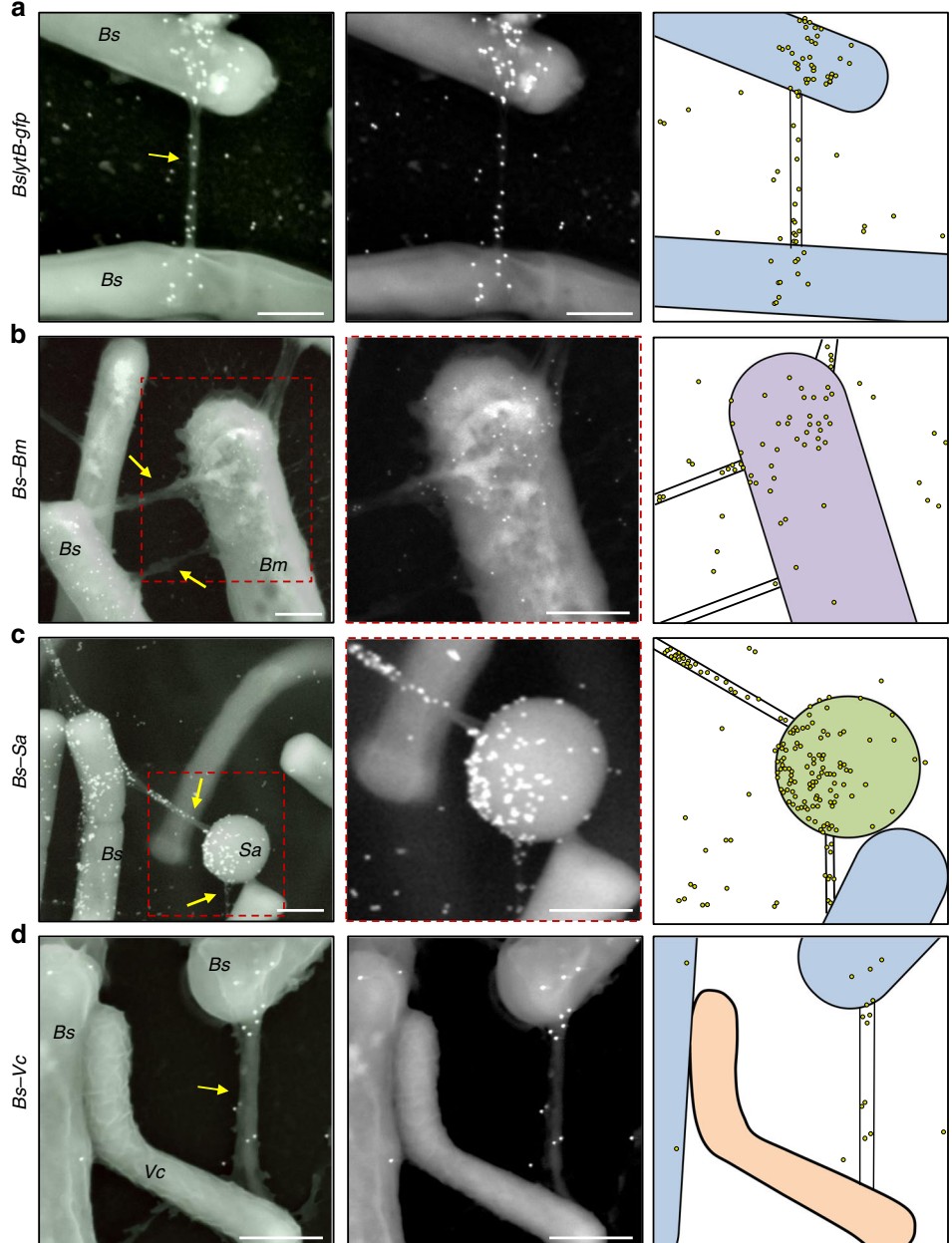

**Fig. 3 LytB molecules travel along intercellular nanotubes to reach the recipient cell surface. a** *Bs* cells expressing LytB-GFP (AB144: *lytB-gfp, ΔymdB, amyE*::P*hyper-spank*-*ymdB, Δhag*) were spotted onto EM grids and subjected to immuno-gold XHR-SEM analysis, using primary antibodies against GFP and secondary gold-conjugated antibodies. Samples were not coated before observation. Shown is an overlay of XHR-SEM images acquired using TLD-SE (through lens detector-secondary electron) for nanotube visualization and vCD (low-kV high-contrast detector) for gold particle (white dots) detection (left panel), and the corresponding vCD image (middle panel). Schematic depicts the interpretive cell layout and highlights the region with gold signal captured by XHR-SEM (right panel). **b** *Bs* cells expressing LytB-GFP (AB170: *lytB-gfp, ΔymdB, amyE*::P*hyper-spank*-*ymdB, Δhag, ΔwapA*) were mixed with *Bm* (OS2, lacking GFP), spotted onto EM grids and subjected to immuno-gold XHR-SEM as described in **a**. Shown is an overlay of XHR-SEM images acquired using TLD-SE for nanotube visualization and vCD for gold particle (white dots) detection (left panel), and a corresponding vCD image of the boxed region (middle panel). Schematic depicts the interpretive cell layout and highlights the region with gold signal captured by XHR-SEM (right panel). **c** *Bs* cells expressing GFP tagged LytB (AB144: *lytB-gfp, ΔymdB, amyE*::P*hyper-spank*-*ymdB, Δhag*) were mixed with *Sa* (MRSA, lacking GFP), spotted onto EM grids and subjected to immuno-gold XHR-SEM as described in **a**. Shown is an overlay of XHR-SEM images acquired using TLD-SE for nanotube visualization and vCD for gold particle (white dots) detection (left panel), and a corresponding vCD image of the boxed region (middle panel). Schematic depicts the interpretive cell layout and highlights the region with gold signal captured by XHR-SEM (right panel). **d** *Bs* cells expressing GFP tagged LytB (AB144: *lytB-gfp, ΔymdB, amyE*::P*hyper-spank*-*ymdB, Δhag*) were mixed with *Vc* (N16961, lacking GFP), spotted onto EM grids and subjected to immuno-gold XHR-SEM analysis as described in **a**. Shown is an overlay of XHR-SEM images acquired using TLD-SE for nanotube visualization and vCD for gold particle (white dots) detection (left panel), and the corresponding vCD image (middle panel). Schematic depicts the interpretive cell layout and highlights the region with gold signal captured by XHR-SEM (right panel). Each experiment was repeated at least five times independently with similar results. Scale bars represent 500 nm. Arrows indicate intercellular nanotubes.

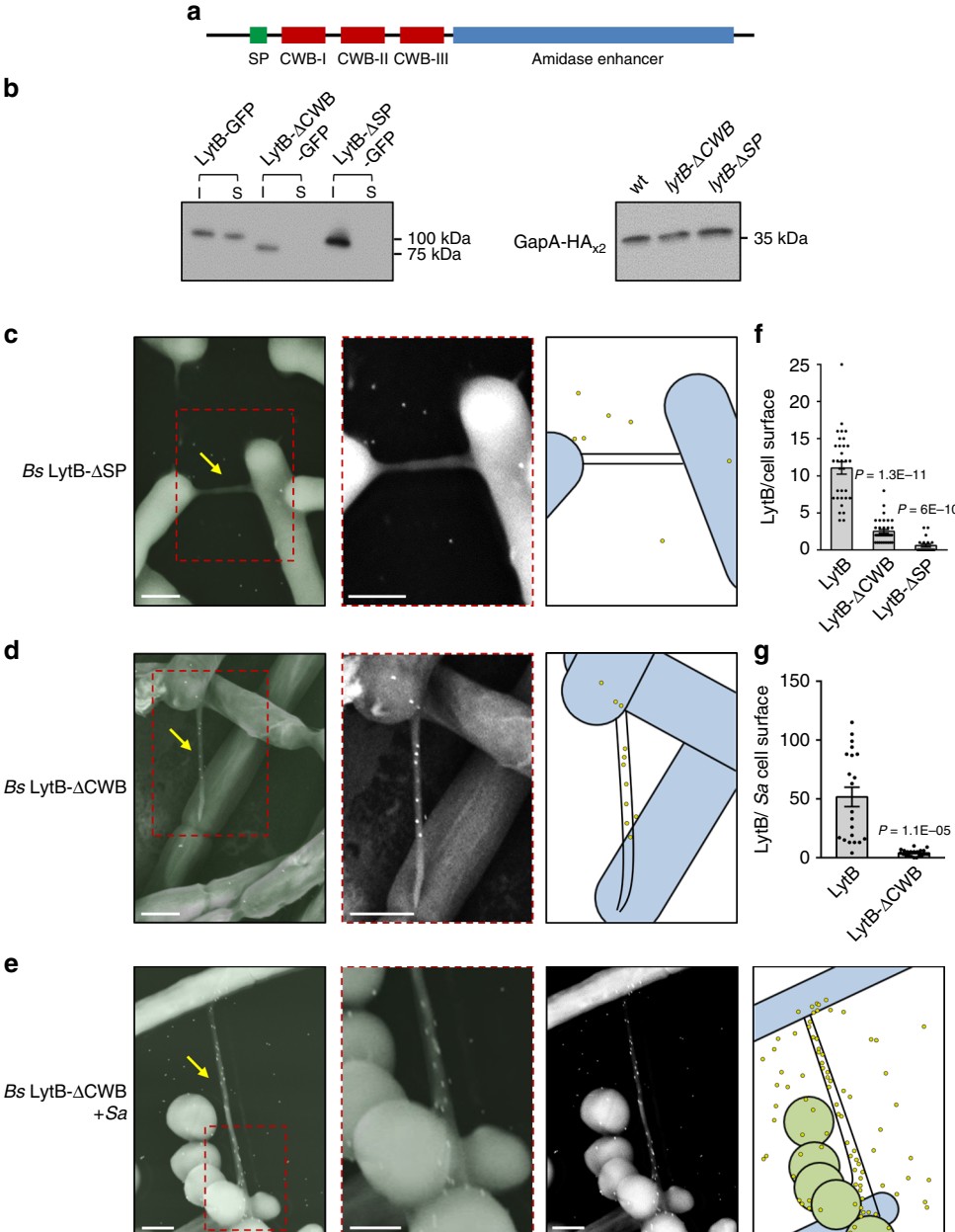

intact *Sa* cells (Fig. 5c, d). Taken together, these results show that LytB-*Bs* and LytC-*Bs* can act from the outside to degrade the CW of various species.

**LytB and LytC from diverse species can couple to facilitate nanotube penetration.** *Bm* harbors homologs of both LytB-*Bs* (LytB-*Bm*) and LytC-*Bs* (LytC-*Bm*), but *Sa* contains only a LytC-*Bs* homolog (LytC-*Sa*) (Supplementary Fig. 13C). We next sought to explore whether pairing of LytB and LytC from diverse species could synergize to facilitate CW cleavage. Therefore, we monitored the degradation activity of LytB-*Bs* on isolated *Bs* sacculi when combined with LytC-*Bm* or LytC-*Sa*. RBB and XHR-SEM analyses showed a significant synergy between LytB-*Bs* and LytC-*Bm* that could not be detected with LytC-*Sa* (Fig. 6a; Supplementary Fig. 13D). Consistent results were obtained using the PI penetration assay with *Bs*, *Bm*, or *Sa* cells (Fig. 6b; Supplementary Fig. 13E, F). The strong synergy between LytB-*Bs* and LytC-*Bm* raises the likelihood that these two enzymes can collaborate in

nature to degrade the bacterium CW and facilitate nanotube penetration. To mimic such a scenario, we treated wt *Bm* or mutated *Bm* lacking *lytC* with purified LytB-*Bs*, and tested the ability of the cells to acquire PI. A sharp decrease in the PI uptake was seen for *Bm* Δ*lytC* in comparison to *Bm* wt (Fig. 6c). In accord, PI uptake was decreased in *Bs* cells deficient for *lytC* treated in a similar manner (Fig. 6c). These results support the view that LytB delivered via nanotubes can interact with the recipient's native LytC to locally facilitate CW degradation and hence nanotube invasion.

We have previously shown that *Bs* kills *Bm* cells grown in proximity by delivering the tRNase toxin WapA via nanotubes[6]. We thus examined the ability of *Bs* to inhibit *Bm* growth, as a readout for establishment of interspecies nanotubes. As expected, *Bs* lacking both *lytB* and *lytC* was impaired in killing *Bm*, most likely due to the reduced number of nanotubes (Figs. 1 and 7a; Supplementary Fig. 14A). Nevertheless, *Bs* lacking solely *lytC* was capable of efficiently killing its *Bm* opponent, an activity that was dramatically decreased when the recipient *Bm* was lacking its

**Fig. 4 CWB domains are not required for LytB localization to nanotubes. a** A schematic model illustrating the N-terminal SP site, three CWB domains, and amidase enhancer region of LytB, based on NCBI BLAST CD-search, KEGG GenomeNet database MOTIF, and EMBL-EBI InterPro. **b** Western blot analysis of intracellular and cell surface levels of LytB fused to GFP from *Bs* wt (AB278: *lytB-gfp, gapA-HA$_{x2}$*), ΔCWB (AB284: *lytB-ΔCWB-gfp, gapA-HA$_{x2}$*), and ΔSP (AB285: *lytB-ΔSP-gfp, gapA-HA$_{x2}$*) strains. Initially, surface proteins were isolated using 1.5 M LiCl from equal number of cells from each strain, followed by intracellular protein isolation by treating the cells with lysozyme and lysis buffer. Shown is a western blot analysis utilizing anti-GFP antibodies to indicate the levels of intracellular (I) and surface (S) LytB (left panel). Intracellular GapA-HA$_{x2}$ levels, detected on a separate western blot with anti-HA antibodies, serve as a loading control (right panel). The experiments were repeated at least three times independently with similar results. **c, d** *Bs* cells expressing GFP tagged LytB-ΔSP (AB263: *lytB-ΔSP-gfp*, Δ*ymdB*, *amyE::*P$_{hyper-spank}$-*ymdB*, Δ*hag*) (**c**) or GFP tagged LytB-ΔCWB (AB250: *lytB-ΔCWB-gfp*, *amyE::*P$_{hyper-spank}$-*ymdB*, Δ*hag*) (**d**) were spotted onto EM grids and subjected to immuno-gold XHR-SEM analysis, using primary antibodies against GFP and secondary gold-conjugated antibodies. Samples were not coated before observation. Shown is an overlay of XHR-SEM images acquired using TLD-SE for nanotube visualization and vCD for gold particle (white dots) detection (left panels), and the corresponding vCD image of the boxed region (middle panels). Schematics depict the interpretive cell layout and highlight regions with gold signal captured by XHR-SEM (right panels). Scale bars represent 500 nm. Arrows indicate the intercellular nanotubes. **e** *Bs* cells expressing GFP tagged LytB-ΔCWB (AB250) were mixed with *Sa* (MRSA, lacking GFP), spotted onto EM grids and subjected to immuno-gold XHR-SEM as described in **c, d**. Shown are (left to right): an overlay of XHR-SEM images acquired using TLD-SE for nanotube visualization and vCD for gold particle (white dots) detection, an enlargement of the boxed region displayed in the left panel, a corresponding vCD image of the left panel, and a schematic depicting the interpretive cell layout, highlighting the region with gold signal captured by XHR-SEM. Scale bars represent 500 nm. Arrows indicate the intercellular nanotubes. **f** Quantitation of cell surface display of the number of gold-labeled LytB molecules per cell in wt (AB144), ΔCWB (AB250), and ΔSP (AB263) strains following immuno XHR-SEM as described in **c, d**. Shown are mean ± SEM and *P* values (unpaired Student's *t* test) of at least three independent experiments ($n_{cells}$ = 50). **g** Quantitation of the number of gold labeled LytB molecules displayed per *Sa* cell surface when mixed with wt (AB144) or ΔCWB (AB250) *Bs* strains following immuno XHR-SEM analysis as described in **e**. Shown are mean ± SEM and *P* values (unpaired Student's *t* test) of at least three independent experiments ($n_{cells}$ = 30). Source data are provided as a Source Data file.

---

endogenous *lytC* gene (Fig. 7b; Supplementary Fig. 14A). Furthermore, a clear reduction in interspecies nanotubes was monitored when *Bs* Δ*lytC* was mixed with *Bm* Δ*lytC* (Supplementary Fig. 14B). However, absence of LytB from *Bs*, *Bm*, or both had no effect on elimination of *Bm* (Fig. 7c; Supplementary Fig. 14A). Together, these results reinforce the view that compatibility between LytB and LytC can determine the multitude of interspecies interaction.

## Discussion

Bacteria produce various apparatuses to conduct intercellular molecular exchange, including transfer of DNA, toxin delivery, as well as nutrient acquisition[6,9,32,33]. However, it remains mysterious how an organelle that physically bridges two adjacent cells is being established to execute these functions. In line with this question, the tactics by which the thick bacterial CW is invaded to pave the way for an intercellular device is yet unknown. Here we report that CW hydrolases can be utilized by the bacterium not only for the emergence of an intercellular nanotube device from the producer cell surface, but also to penetrate the neighboring recipient to establish an intercellular route. We show that the Gram-positive bacterium *Bs* uses the CW remodeling enzymes, LytC-amidase, and its enhancer LytB, to cross the donor and the recipient CW barriers. Our data suggest that LytB, and to a lesser extent LytC, migrate along nanotubes and reach the surface of neighboring bacteria. Remarkably, such delivery of LytB-*Bs* molecules could occur in an interspecies manner, as LytB-*Bs* molecules were deposited over the surface of foreign species in proximity to nanotube penetration sites, suggesting that they act to locally remodel the recipient CW to enable nanotube access. Such delivery of CW hydrolases along with the synergy observed between interspecies pairs of LytB and LytC insinuate that nanotube penetration through the CW might be dependent on interspecies compatibility between these hydrolases, among other enzymes with a similar function. This could be a crucial parameter in determining the extent of nanotube-mediated cytoplasmic sharing in heterogenous bacterial communities.

The LytC amidase hydrolyses the highly conserved amide bond present between the *N*-acetyl muramic acid residue and the L-alanine of the PG backbone, indicating that LytB and LytC pairs potentially have the capacity to modify the CW of multiple species, yet the efficiency of their delivery could vary. The observation that LytB lacking its CW binding domains can still localize to nanotubes, but was rarely observed over the cell surface, raises the possibility that LytB molecules are transported through the membranous nanotubes in a SP domain-dependent manner, and employ the CW binding domains to fasten to the recipient surface. This possibility is strengthened by the findings that LytB-*Bs* molecules failed to be deposited onto the outer membrane surface of the Gram-negative *Vc*. Henceforth, our data indicate that CW lytic enzymes could be a preferential mechanism for interactions among Gram-positive species, as such they are likely to be one of the earliest nanotube-delivered cargo. It is tempting to assume that following CW perturbation of a nanotube recipient Gram-positive bacterium, a membrane fusion event is likely to take place. This could be also applicable to interactions involving Gram-negative bacteria or even between bacteria and their mammalian host cells.

Utilization of CW hydrolases for the emergence of extracellular appendages from the producer cell has been documented in various bacterial species, for example the EtgA from *E. coli* in type 3 secretion system[34], FlgJ from *Salmonella* in flagellum[35], VirB1 from *Agrobacterium tumefaciens* in type 4 secretion system[36], EssH from *Sa* in type 7 secretion system[37], TraG from *Enterococcus faecalis*, and CwlT from *Bs* in type 4 conjugation pili[38,39]. Likewise, prophage-encoded endolysins are capable of poking holes within the PG layer, leading to membrane vesicle release by the host[40]. Intriguingly, *Anabaena sp.* utilizes a CW hydrolase to assemble an array of nanopores on the bacterial division septum to connect daughter bacteria residing within the same chain[41]. Notably, in all of these mentioned cases, CW hydrolases were found to assist the formation of extracellular structures by the producing bacterium. Similarly, we observed that the lack of LytB and LytC reduces the emergence of nanotubes from the generating bacterium. However, the requirement for these hydrolases to penetrate the CW of the nearby bacterium highlights another dimension for their functionality in nature. It is tempting to speculate that additional intercellular organelles, such as conjugation systems, cross the recipient CW with the aid of dedicated CW hydrolases.

In light of our findings, we propose that Lyt proteins act as keys that enable penetration of nanotubes to recipient bacteria, with the recipient's own array of CW lytic protein reservoir

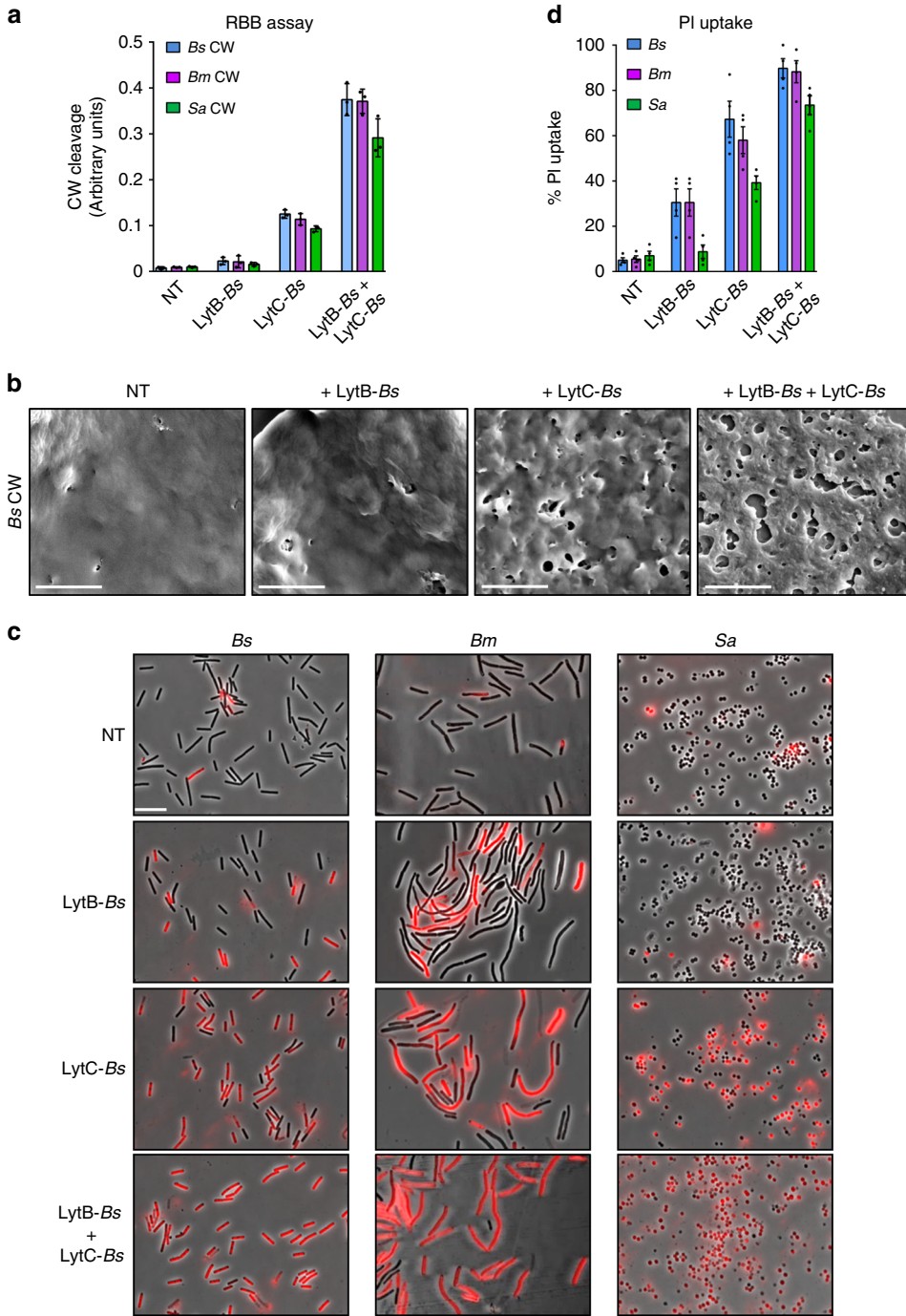

**Fig. 5 LytB-*Bs* and LytC-*Bs* can exogenously cleave CW of foreign species. a** CWs were isolated from *Bs* (PY79), *Bm* (OS2), or *Sa* (MRSA), labeled with RBB, mixed with purified LytB-*Bs*, LytC-*Bs* or both. The cleaved products were separated by centrifugation and absorbance of supernatant was measured at 595 nm. Shown are mean ± SEM of at least three independent experiments. **b** *Bs* sacculi derived from PY79 cells were incubated for 30 min with purified LytB-*Bs*, LytC-*Bs* or both, spotted onto poly ʟ-lysine coated coverslips and visualized by XHR-SEM. Scale bars represent 500 nm. **c** *Bs* (PY79), *Bm* (OS2) or *Sa* (MRSA) cells were incubated for 10 min with purified LytB-*Bs*, LytC-*Bs* or both, stained with PI and visualized by fluorescence microscopy. Shown are overlay images of phase contrast (gray) and fluorescence from PI staining (red). Scale bar represents 5 µm. **d** Quantitation of PI uptake by *Bs* (PY79), *Bm* (OS2), or *Sa* (MRSA) cells treated as described in **c**. Shown is the percentage of PI-labeled cells and mean ± SEM of at least three independent experiments ($n_{cells} = 400$). NT indicates no treatment (samples contained only buffer). The experiments were repeated at least three times independently with similar results. Source data are provided as a Source Data file.

contributing to the efficiency of this interaction. Given that nanotubes may be used for both cooperation and competition, our data suggest that bacteria might exploit Lyt compatibility to selectively design their dwelling niche, by physically connecting and exchanging cytoplasmic molecules preferentially with favored species. In line with this view, bacteria might develop resistance mechanisms for such connections to defend from potential delivered toxins and to protect their nutritional resources from being grasped by competing bacteria.

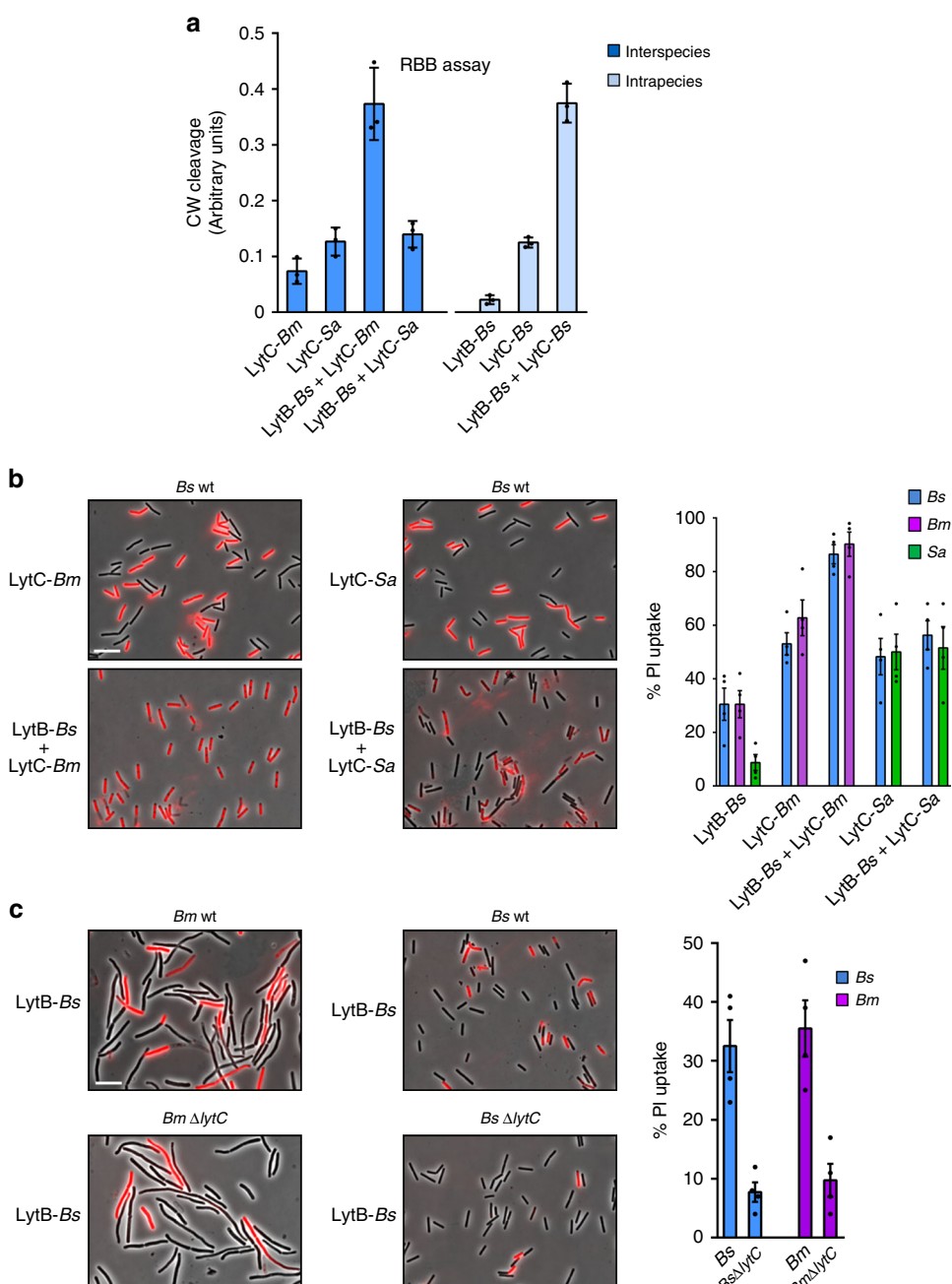

**Fig. 6 Interspecies LytB and LytC can pair to enhance CW degradation. a** CW was isolated from *Bs* (PY79), labeled with RBB and mixed with purified proteins in the various indicated combinations. The cleaved products were separated by centrifugation and absorbance of supernatant was measured at 595 nm. Shown are mean ± SEM of at least three independent experiments. **b** *Bs* wt (PY79) cells were incubated for 10 min with purified LytB-*Bs* with or without LytC from *Bm* or LytC from *Sa*, stained with PI, and visualized by fluorescence microscopy. Shown are overlay images of phase contrast (gray) and fluorescence from PI staining (red). Scale bar represents 5 μm. Graph presents quantitation of PI uptake by the indicated cells following the above treatments. Shown is the percentage of PI-labeled cells and mean ± SEM of at least three independent experiments ($n_{cells} = 400$). **c** *Bm* wt (OS2) or Δ*lytC* (ABm2) and *Bs* wt (PY79) or *Bs* Δ*lytC* (AB3) strains were incubated for 10 min with LytB-*Bs*, stained with PI and visualized by fluorescence microscopy. Shown are overlay images of phase contrast (gray) and fluorescence from PI staining (red). Scale bar represents 5 μm. Graph presents quantitation of PI uptake by the indicated cells following the above treatments. Shown is the percentage of PI-labeled cells and mean ± SEM of at least three independent experiments ($n_{cells} = 400$). Source data are provided as a Source Data file.

## Methods

**Bacterial strains and general methods**. *Bs* are derivatives of the wild-type strain PY79, *Bm* is a wild-type soil isolate (OS2) and its derivatives, *Sa* is MRSA strain and *Vc* is O1 El Tor N16961 strain. Bacterial strains and plasmids are listed in Supplementary Data 1, and primers are listed in Supplementary Data 2. All general methods were carried out as described previously[42]. Transformation into *Bm* OS2 cells was conducted as described[43]. In brief, *Bm* OS2 cells were grown up to 1.0 $OD_{600}$, washed with electroporation buffer (25% PEG 6000 and 0.1 M sorbitol) and

resuspended in 1 ml of electroporation buffer. Electroporation was carried out with 0.1 ml of cells supplemented with 500 ng of linear DNA at 1500 V (Bio-Rad). Cells were then diluted into 1 ml of LB and plated on selective antibiotic plate.

**Site-directed mutagenesis**. Site-directed mutagenesis was carried out to introduce point mutations into the LytC amidase active sites. Four amino acid residues, critical for the amidase activity of LytC (histidine-328 and 395, glutamic acid-342 and 461),

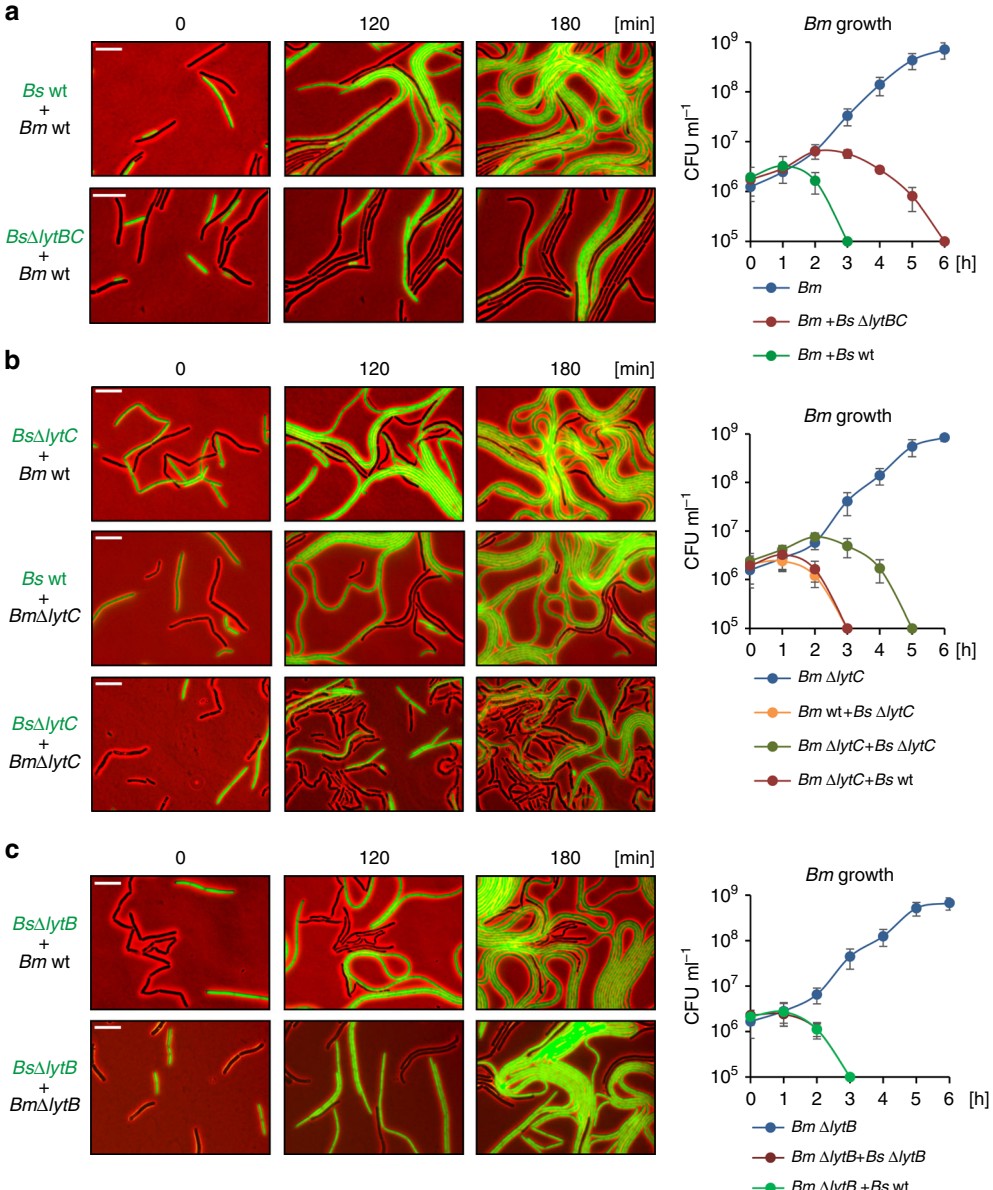

**Fig. 7 Inhibition of *Bm* by *Bs* is mediated by interspecies pairing of LytB and LytC. a** *Bm* wt (OS2) was mixed with GFP expressing *Bs* wt (AR16: P*rrnE*-*gfp*) or *Bs* Δ*lytBC* (AB264: Δ*lytBC*, P*rrnE*-*gfp*) and subjected to time lapse microscopy. Shown are overlay fluorescence from GFP (green) and phase contrast (red) images, captured at the indicated time points. *Bm* cells are shown in black while *Bs* cells are shown in green. Graph presents colony-forming unit (CFU/mL) of *Bm* cells grown alone or in a mixture with the indicated *Bs* strains. Each time point represents an average value and SD of three independent experiments. **b** *Bm* wt (OS2) or *Bm* Δ*lytC* (ABm2) were mixed with GFP expressing *Bs* wt (AR16: P*rrnE*-*gfp*) or *Bs* Δ*lytC* (AB227: Δ*lytC*, P*rrnE*-*gfp*) and subjected to time lapse microscopy as presented in **a**. Graph presents colony forming unit (CFU/mL) of *Bm* cells grown alone or in a mixture with the indicated *Bs* strains. Each time point represents an average value and SD of three independent experiments. **c** *Bm* wt (OS2) or *Bm* Δ*lytB* (ABm1) were mixed with GFP expressing *Bs* Δ*lytB* (AB265: Δ*lytB*, P*rrnE*-*gfp*) and subjected to time lapse microscopy as presented in **a**. Graph presents colony forming unit (CFU/mL) of *Bm* cells grown alone or in a mixture with the indicated *Bs* strains. Each time point represents an average value and SD of three independent experiments. Time lapse microscopy experiments were repeated at least three times independently with similar results. Scale bars represent 5 μm. Source data are provided as a Source Data file.

were selected following NCBI Conserved Domain annotation, and as described previously for the *Bs* sporulation amidase CwlC, which is homologous to LytC[44]. The histidine (H) residues were replaced by leucine (L), and glutamic acid (E) residues were replaced by glutamine (Q), as these replacements have most severe effect on CwlC amidase activity[44]. For mutagenesis, DNA fragments were amplified using primers LytC-SDM (P1-P10) that are listed in Supplementary Data 2. The fragments were then joined using Gibson assembly mix and transformed into the indicated *Bs* strains. The point mutations were confirmed by DNA sequencing.

**Molecular exchange assay.** For detecting molecular exchange, antibiotic transfer assay was carried out as described previously[3,12] and in the legend to Supplementary Fig. 2. Donor and recipient strains used for the molecular exchange assay

are listed in Supplementary Data 1. Respective donor and recipient strains were grown to mid logarithmic phase, after which cells were mixed in 1:1 ratio ($OD_{600}$ = 0.8 or 0.08) and incubated in LB supplemented with 1 mM IPTG for 4 h at 37 °C with gentle shaking. Equal numbers of cells were spotted onto either double selective LB agar plates containing Chloramphenicol and Kanamycin for detection of protein exchange or triple selective LB agar plates containing Chloramphenicol, Kanamycin, and Lincomycin for detection of plasmid exchange and incubated overnight at 37 °C. Cells were spotted onto LB agar plates as a control. Fluorescent colonies were detected using Bio-Rad ChemiDoc MP Imaging System.

**Bacterial competition assay.** Overnight cultures of *Bs* and *Bm* were diluted to $OD_{600}$ 0.1, mixed and mounted onto a metal ring (A-7816, Invitrogen) filled with

LB agarose (1.5%). Cells were incubated in a temperature-controlled chamber at 30 °C, and followed by time lapse fluorescence microscopy, utilizing Eclipse Ti (Nikon) or Axio Observer Z1 (Zeiss) microscopes. Images were captured using Photometrics CoolSNAP HQ2 (Photometrics, Roper Scientific, USA) and image processing was performed using NIS Elements AR version 4.5 (Nikon) or Metamorph 7.4 (Molecular Devices).

**Immunofluorescence microscopy**. Mid logarithmic phase *Bs* cells centrifuged, inoculated into the fresh LB at O.D.600 0.1 and incubated for 4 h at 37 °C with gentle shaking. Cells were harvested, spotted onto poly L-lysine coated coverslips and incubated for 10 min. The excess liquid was removed, and the cells were treated with 2% BSA for 10 min. The cells were then treated with anti-GFP primary antibodies (1:1000) for 30 min, washed six times with 1xPBS, and subsequently treated with FITC-conjugated secondary antibodies (1:1000). Cells were then washed six times with 1X PBS and then mounted on glass slides using Immumount (Thermo Scientific) and visualized by Eclipse Ti (Nikon, Japan), equipped with CoolSNAP HQ2 camera (Photometrics, Roper Scientific, USA). System control and image processing were performed with NIS Elements AR 4.3 (Nikon, Japan).

**XHR-SEM analysis**. *Bs*, *Bm*, *Sa*, or *Vc* cells were spotted onto EM grids and incubated on LB agar plates for 4 h. Grid-attached cells were washed three times with PBS×1, fixed with 2% paraformaldehyde, and 0.01% glutaraldehyde in sodium cacodylate buffer (0.1 M, pH 7.2) for 10 min at 25 °C. Cells were then left overnight for fixation in 2% glutaraldehyde in sodium cacodylate buffer (0.1 M, pH 7.2) at 4 °C. To dehydrate the cells, the EM grids underwent a series of washes in increasing concentrations of ethanol (25, 50, 75, and 96%) and kept overnight in vacuum. Samples were coated and observed using Through-Lens Detector operated at Secondary Electron (TLD-SE) mode by Magellan XHR-SEM (FEI).

**Immuno XHR-SEM analysis**. *Bs*, *Bm*, *Sa*, or *Vc* cells alone or in mixture explained in figure legends, were grown on EM grids (mesh copper grids, EMS), and grid-attached cells were washed three times with PBS × 1, fixed with 2% paraformaldehyde and 0.01% glutaraldehyde for 10 min at 25 °C. Subsequently, grids were washed three times in PBS × 1 and were incubated in PBS × 1 containing 2% BSA and 0.1% Tween 20 for 30 min at 25 °C and washed twice with PBS × 1. Next, grids were incubated for 2 h at 25 °C with rabbit anti-GFP or anti-HA antibodies (Thermo Scientific, USA) diluted 1:200 in PBSx1 containing 1% BSA. Grids were then washed three times with PBS × 1 and incubated for 1 h at 25 °C with gold-conjugated goat anti-rabbit antibodies (Jackson, USA) diluted 1:100 in PBS × 1. Grids were washed three times with PBS × 1 and fixed with 2.5% glutaraldehyde in sodium cacodylate buffer (0.1 M, pH 7.2) for 1 h at 25 °C. Grids were then washed gently with water, and cells were dehydrated by exposure to a graded series of ethanol washes [(25, 50, 75, 95, and 100%) ×2; 10 min each], followed by overnight incubation in vacuum. Specimens were imaged without coating by Magellan XHR-SEM (FEI) using TLD-SE and vCD modes. At first, the image of cells and nanotube was acquired in secondary electron mode using TLD-SE and then the same field was imaged using vCD (low-kV, High contrast) detector.

**Quantitation of LytB molecules on *Sa* cell surface**. Gold particles denoting the LytB molecules on *Sa* cell surface were quantitated by dividing the cell into two halves- proximal and distal. The proximal half contained the nanotube attachment site and the half opposing the nanotube attachment site was designated as distal. Gold particles were counted and the ratio between proximal and distal side was plotted. The background signal was estimated in individual images by counting the particles in the cell-free regions from multiple areas that were equal in size to *Sa* cells.

**Protein extraction and western blot analysis**. *Bs* cells at mid logarithmic phase were centrifuged, inoculated into fresh LB at O.D.600 0.1 and incubated for 4 h in 37 °C with gentle shaking. Cell surface proteins were isolated as described previously[25]. Briefly, cells were harvested and washed with 10 mM Tris-HCl, pH 8.0. Cell pellet was then resuspended in buffer containing 1.5 M LiCl, 25 mM Tris-HCl at pH 8.0, and kept in ice for 10 min. The suspension was then centrifuged and supernatant containing the protein was collected. The proteins were precipitated using 10% TCA (w/v). The remaining cell pellet was then used to isolate the intracellular proteins following treatment with lysozyme (20 μg ml⁻¹) and subsequently with lysis buffer (150 mM NaCl, 1% Triton X-100, 50 mM Tris pH 8.0).

Next, proteins were separated by SDS-PAGE on 4–15% polyacrylamide gradient gels (Bio-Rad). Proteins were then electroblotted onto the Immobilon-P membranes (Millipore), and the membrane was blocked for 1 h with 5% skim milk in TBSx1 (50 mM Tris-Cl, pH 7.5, 150 mM NaCl) and 0.5% Tween-20. Blots were then probed with polyclonal rabbit anti-GFP or rabbit anti-HA antibodies (1:5000 in 0.05% Tween-20, 5% skim milk in TBSx1) followed by peroxidase conjugated goat anti-rabbit secondary antibody (Bio-Rad) (1:10,000 in 0.05% Tween-20, 5% skim milk). The signal was detected using EZ-ECL kit (Biological Industries, Beit Haemek, Israel).

**LytB and LytC expression and purification**. Proteins were expressed from the plasmid pQE32 containing *lytB* or *lytC* in *E. coli* BL21 (DE3). Cells were grown in LB supplemented with ampicillin (50 mg ml⁻¹) up to OD600 0.5 and induced by the addition of 0.5 mM IPTG for 16 h at 16 °C. Purified proteins were extracted according to Qiagen protocol (The QIAexpressionist). Cells were harvested by centrifugation and resuspended in lysis buffer (50 mM NaH₂PO₄, 300 mM NaCl, 10 mM imidazole at pH 8.0). Next, lysozyme (1 mg ml⁻¹), RNaseA (10 μg ml⁻¹), DNase I (5 μg ml⁻¹) were added and cells were incubated in ice for 30 min and treated using Fastprep (FastPrep (MP) 6.5, 30 s, ×3) and the lysates were collected after centrifugation at 10,000 × *g* for 20 min. The lysates were then mixed with Ni-NTA slurry (Qiagen) and incubated at 4 °C for 1 h with gentle mixing. The mixture was then poured into Poly-Prep chromatography column (Bio-Rad) and washed three times with wash buffer (50 mM NaH2PO4, 300 mM NaCl, 20 mM imidazole, pH 8.0) and eluted in elution buffer (50 mM NaH₂PO₄, 300 mM NaCl, 250 mM imidazole, pH 8.0). The eluted fractions were pooled and used for investigation.

**CW purification and RBB labeling**. CW preparation from the *Bs*, *Bm*, or *Sa* strains was based on McPherson and Popham[45]. Cells at mid logarithmic phase grown in LB at 37˚C were harvested and boiled in 4% SDS solution for 30 min. The solution was then centrifuged at 15,000 × *g* for 15 min and pellet was washed ×10 times with ddH₂O to remove the SDS. Isolated CW were treated with 10 mg ml⁻¹ DNase I and 50 mg ml⁻¹ RNase A for 2 h at 37 °C in Tris buffer (100 mM Tris HCl at pH 7.5, 20 mM MgSO₄), followed by an overnight incubation with 100 mg ml⁻¹ proteinase K at 37 °C in Tris buffer supplemented with 10 mM CaCl₂. CW was boiled again in 1% SDS for 15 min, and centrifuged at 12,000 × *g* for 10 min at room temperature. The pelleted CW was then washed thrice with ddH₂O, once in of 8 M LiCl, and twice again in ddH₂O. CW was then incubated with 20 mM RBB (Sigma) in 0.25 M NaOH overnight at 37 °C. The preparation was neutralized with HCl, and the dye-labeled CW was pelleted by centrifugation (15,000 × *g*, 30 min, room temperature). Extracted CW was then washed repeatedly with ddH₂O until the supernatant was clear. The final pellet was resuspended in 2 ml ddH₂O and stored at 4 °C.

**CW cleavage assay**. RBB-labeled CW isolated from 50 ml cell culture was treated with 10 μM ml⁻¹ of each indicated purified proteins, both when added alone or in mixture, and incubated at 37 °C for 30 min. Samples were then incubated at 95 °C for 5 min to stop the reaction, and subsequently centrifuged at 15,000 × *g* for 30 min. The supernatant, containing the soluble cleavage products, were collected and absorbance was measured at 595 nm.

**CW cleavage visualization by XHR-SEM**. *Bs* intact cells or the isolated CW was treated with the 10 μM ml⁻¹ of each indicated purified proteins, both when added alone or in mixture, and incubated at 37 °C for 30 min. Samples were then incubated at 95 °C for 5 min to stop the reaction, and subsequently spotted onto poly L-lysine coated coverslips, fixed with 2% paraformaldehyde and 0.01% glutaraldehyde for 10 min, 2.5% glutaraldehyde in sodium cacodylate buffer (0.1 M, pH 7.2) for 1 h at 25 °C. Coverslips were then washed gently with water and were dehydrated by exposure to a graded series of ethanol washes [(25, 50, 75, 95, and 100%) ×2; 10 min each], followed by overnight incubation in vacuum. Samples were coated and observed in Magellan XHR-SEM (FEI).

**Propidium iodide staining**. *Bs*, *Bm*, or *Sa* cells at mid logarithmic phase were harvested, washed in PBSx1, and suspended in PBSx0.3 to impose hypotonic stress. Cells were then treated with 10 mM ml⁻¹, both when added alone or in mixture, of each LytB, LytC, or both at room temperature for 10 min. Cells were then treated with 5 μg ml⁻¹ PI for 2 min and visualized by Eclipse Ti (Nikon, Japan), equipped with CoolSnap HQII camera (Photometrics, Roper Scientific, USA). System control and image processing were performed with NIS Elements AR 4.3 (Nikon, Japan).

**Statistical analysis**. Statistical analysis was performed using the MS office excel and GraphPad Prism softwares. *P* values obtained are indicated in the respective figures.

**Reporting summary**. Further information on research design is available in the Nature Research Reporting Summary linked to this article.

## Data availability
The source data for Figs. 1c, d, 2b, 4b, f, g, 5a, d, 6 a–c, 7 a–c and Supplementary Figs. 1B, 2A, 4B–D, 7C, D, 9B, 10B, 11D, 13B, and 14A, B are provided as a Source data file. All data supporting the findings of this study are available from the corresponding author upon request.

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

## Acknowledgements

We thank E. Blayvas and A. Ben-Hur (Hebrew University, Israel) for help with XHR-SEM. We are grateful to A. Rouvinski (Hebrew University, Israel) and members of the Ben-Yehuda and Rosenshine laboratories for valuable discussions. This work was supported by the European Research Council (ERC) Advanced Grant (339984), and the NSF/BSF-United States-Israel Binational Science Foundation (2017672) awarded to S.B.-Y., and the ERC Synergy grant (810186) awarded to S.B.-Y. and I.R.

## Author contributions

A.K.B. performed the experiments. A.K.B., S.B.-Y, and I.R. conceived the experiments and analyzed the data. A.K.B., S.B.-Y., and I.R. wrote the manuscript. S.B.-Y. and I.R. managed the project.

## Competing interests

The authors declare no competing interests.
