## [Peer Review File · Nature Communications]

Reviewers' comments:

Reviewer #1 (Remarks to the Author):

The manuscript Donor-delivered cell wall hydrolases facilitate nanotube penetration to recipient bacteria is a potentially interesting work with great impact in how we currently understand bacterial cell-to-cell communication in monoclonal or mixed communities. A lot of experiments are presented in the manuscript to support the authors' Results are introduced in a smoothly in such the story is easy to follow. I have a few comments that I hope will help the authors to improve the manuscript.

In all figures: It would be nice to present the graphics with statistical analyses, to see whether the differences detected in WT and mutants are significant.

Lyt proteins migrate along the nanotubes and reach the surface of the neighbouring cell. Lyt proteins degrade CW and facilitate the connection of the nanotube with the neighbouring cell. If this is the case, one should expect the formation of nanotubes in the membrane regions where LytB localizes (Fig. 2 shows the puncta localisation of LytB). Do the authors know if this is the case?

Is there a negative control of a protein that is not transferred via nanotubes? What about the order of proteins that are transferred? Based on the results of this paper, LytB should be one of the first enzymes to be transferred by nanotubes, otherwise the cell-to-cell connection cannot be made.

At the beginning of the results section, the authors mentioned that they tested the role of Lyt proteins in nanotube formation. I miss here a logic rationale that led the authors to follow up these enzymes.

The signal in the vCD image in the right column of the figures sometimes does not correspond to the signal that is in the real microscopy images. In general, the localization of the signal is much clearer in the microscopy images than in the vCD image. Some readjustments are needed.

Fig 1, 3 and 4 needs additional data in supplemental material showing an entire EM field of the cells. this way, the reader has a better perception of how efficient was the localization of the signal in the nanotubes

Reviewer #2 (Remarks to the Author):

In the manuscript "Donor-delivered cell wall hydrolases facilitate nanotube penetration to recipient bacteria", Baidya et al. describe a putative mechanism of how bacterial nanotubes deliver cargo to the cytoplasm of recipient cells via controlled cell wall breaches.

If correct, this mechanism would expand our view of bacterial cell-to-cell communication and provide a mechanistic explanation for the formation of the still mysterious nanotubes. This would be interesting to a broad readership. The interplay between LytC and LytB (though somewhat preliminary) is very interesting even without the nanotube context. In essence, I think there are some very interesting threads here, but the decisive experiment (directly demonstrating migration of lytB/C along nanotubes into recipient cells as well as cell wall remodeling in recipient and donor cells) is missing. I also have some concerns about experimental data and their interpretation. Please see my detailed comments below.

1. The conclusions are often overstated. The end of the introduction, for example, states "LytB molecules were found to travel along..." and "can cooperate to promote nanotube penetration". The conclusion of "traveling" would require time-resolved imaging of the dynamics of fluorescently-labelled LytB. Nanotube penetration is not directly demonstrated, but hypothesized from indirect data. The abstract states "the fewer nanotubes of the lytBC mutant were deficient in penetrating" – again, no evidence for nanotube penetration is presented. These conclusions should thus be weakened to indicate that they are still hypotheses/speculation.

2. Fig. S1D. The mutants' deficiency in intercellular trafficking are not clear to me. The only difference I can make out is between Δ lytBC and the rest, but these spots look like dried cell material. I am not sure I would interpret the WT as successful protein exchange. The lighter spot for the Δ lytBC mutant might as well reflect a lower cell density in the spotted mutant volume. Likewise, the plasmid exchange data are very difficult to interpret. Are the small spots really all colonies? Would they be bigger if you incubated the cell for a longer time? Could they simply be abortive growth of a subpopulation due to leftover donor enzyme that degrades antibiotics in the spot (this could be tested by repurifying colonies on selective plates)? Also, in these assays, how can you distinguish between nanotube-mediated transfer and other means of transformation (e.g., DNA uptake from lysed cells). Did they repeat this assay in a competence(-) background?

3. Page 6, line 18/19. What is your explanation for the lack of GFP fluorescence? Did you try any other fluorophores/HALO tags etc.? A fluorescently-labeled LytB would be very useful to prove the main point of the paper.

4. Nanogold labeling in general: E.g., Figure 3 – the schemata often omit background signal. While I understand why this is done, I think you should include background as well, otherwise the schema becomes an idealized version of the actual image, which means a decrease in information. For example, in Fig. 3A, many gold particles can be seen in a and b, but are not drawn in c.

5. I am not sure about the enrichment of gold particles near presumed nanotube attachment sites (e.g. in the Staph aureus experiment, Figure 3C) – could this be quantified somehow (density/random square as a function of distance from the attachment site)?

6. Page 9, line 7 ff., "the truncation abolished the placement of the protein..." – I would not use "abolish" since both the images (Fig. 4C b) and the quantification (Fig. 4 F) do reveal a few LytB particles.

7. Page 9 line 15 – I am not sure the signal on Sa is significantly lessened – it is hard to differentiate background from signal in Fig. 4E c. There are many gold particles in and around the Sa cells here, which are not drawn into the schema in d (see also my point 4 above). The authors might have determined that these are clearly background, but if so, please state in the text how this was determined. In this context, how variable is gold labeling in these assays? Please add statistics to the quantifications.

8. Page 9, line 19 – I would not write "infer" here, which would imply a basis in data or direct observations. The last bit of the sentence includes the statement "...to enable efficient nanotube penetration". Penetration is not shown, so I would use "speculate" or "hypothesize" instead of "infer".

Reviewer #3 (Remarks to the Author):

In this study, the authors are following up on their previous work on the intercellular membranous

nanotubes that form between *Bacillus* and other species. Here, they address the question of how these nanotubes are able to cross the peptidoglycan cell wall of the producer and receiver cells. In particular, the focus is on a pair of proteins present in/on the nanotubes structures, LytB and LytC.

The authors convincingly show that LytBC are important for nanotube formation and that LytB can be transferred between cells. This is a significant finding as it sheds insight into the mechanism of this form of cell-to-cell interaction.

While the main finding of the role of LytB and LytC in tubule formation is solid, I'm very hesitant about the connection with and transfer into diverse species part of their findings.

- First of all, unless I'm overlooking something, the only evidence for the enzymatic activity of LytC that is shown is the need for LytB. Is it possible that LytBC are just "structural" proteins needed for the formation of the nanotubes? While there is precedent for cell wall remodeling to be needed for production of secretion systems/surface structures, is it definitely LytC that is doing it? The authors do show that LytC can cleave cell walls, but does it **need** to in order to form the nanotubes? The authors don't really show this. It would be nice to see that a catalytically dead LytC has the same phenotype as a LytC deletion.

- I'm not really convinced that the nanotubes formed between *Bacillus* and *Vibrio* are actually connecting with *Vibrio* as the text seems to imply. LytB is not being transferred, so how do you know that the connection is actually being made? The picture in fig S4 looks more like membrane fusion (or maybe EM artifacts) than the clearly defined nanotubes that form between the gram-positive organisms. At the very least, I would like to see a wt vs Δ lytBC mutant with Vc to confirm that the connections to Vc like shown in fig S4 go away. Alternatively, maybe the authors could show that something is actually being transferred to the Vc in the context of these lytBC experiments.

- Continuing with that point, I would like to see confirmation whether LytC in the receiver cell is or is not needed for nanotube connection to be made. i.e. KO the LytC from Sa and see if you still get nanotubes + LytB transfer. Or a use a Bs that is Δ ymdB Δ lytBC that doesn't produce nanotubes, can it be a receiver still?

- Related to the previous point, is it possible that the "continual" nanotubes seen in Fig 1, are an indicator of the Δ lytBC cells being unable to act as receiver cells, as much as it is about producing fewer nanotubes? Since the nanotubes are only visible by EM, and their dynamics aren't really visible in real time, is it possible that the reason you see a lot of nanotubes forming in the wt is that they successfully attach to receiver cells and are therefore stabilized. Meanwhile in the Δ lytBC, the nanotubes are less stable so fewer of them are visible.

Response to Reviewers' comments

We thank the Referees for the helpful and constructive comments on our manuscript: "Donor-delivered cell wall hydrolases facilitate nanotube penetration to recipient bacteria". In the revised manuscript, we attempted to address all the concerns raised. Below we provide a summary of our major modifications, followed by a detailed point-by-point response (black text) to the raised comments (blue text).

Summary of the major changes

- 1) We added additional controls to substantiate the specific localization of LytB to nanotubes (Fig S6).
- 2) In the revised version, we provide the original secondary electron mode (TLD-SE) images to clarify that the signal from the vCD mode overlaps with that from the secondary electron mode (Fig. S5, S6, S7, S8, S11).
- 3) In the revised version, full original fields captured by SEM were added. This will provide the reader with a better perception of the efficiency of LytB localization to nanotubes. These additional supplementary images show that the images we present in the main part of the manuscript are indeed representative (Fig S1; Fig S5; Fig S11).
- 4) We provide point mutation analysis showing the requirement of LytC amidase catalytic activity for nanotube formation (Fig S3).
- 5) A detailed analysis of the nanotube-mediated molecular exchange assay is provided (Fig S2; Reviewer's Fig 2).
- 6) Quantification analysis of LytB molecules transferred between species was added (Fig S9).
- 7) Additional support to substantiate the Bs-Vc interaction is provided in the revised version (Fig S10).
- 8) *P* values were added to the graphics throughout the manuscript when needed.
- 9) We toned down our statements and modified the text according to the Reviewers' requests.

Point-by-point response

Reviewer #1 (Remarks to the Author):

The manuscript Donor-delivered cell wall hydrolases facilitate nanotube penetration to recipient bacteria is a potentially interesting work with great impact in how we currently understand bacterial cell-to-cell communication in monoclonal or mixed communities. A lot of experiments are presented in the manuscript to support the authors' Results are introduced in a smoothly in such the story is easy to follow. I have a few comments that I hope will help the authors to improve the manuscript.

1) In all figures: It would be nice to present the graphics with statistical analyses, to see whether the differences detected in WT and mutants are significant.

According to the Reviewer's comment, when relevant, *P* values were added to the graphics throughout the manuscript (e.g. Fig 1; Fig 2; Fig 4).

2) Lyt proteins migrate along the nanotubes and reach the surface of the neighbouring cell. Lyt proteins degrade CW and facilitate the connection of the nanotube with the neighbouring cell. If this is the case, one should expect the formation of nanotubes in the membrane regions where LytB localizes (Fig. 2 shows the puncta localisation of LytB). Do the authors know if this is the case?

Indeed, we observed by immuno XHR-SEM that nanotube emanate from sites containing multiple LytB molecules (e.g. Fig. 3A). These high densities of protein molecules likely correspond to the LytB foci observed on the bacterium surface by immunofluorescence microscopy. This possibility is indicated in the text (p7, lines 15-19).

3) Is there a negative control of a protein that is not transferred via nanotubes?

To address this concern, we performed immuno XHR-SEM utilizing the ribosomal protein RplA fused to GFP or an unrelated surface exposed membrane protein YueB fused to YFP. No gold signal was obtained from the cell harbouring RplA-GFP, and no nanotube-localized signal was detected from YueB-YFP. These results are now included in Fig S6. It should be noted that very few molecules of LytC were detected over nanotubes (Fig S7B-S7C), emphasizing the specificity of LytB localization to nanotubes.

4) What about the order of proteins that are transferred? Based on the results of this paper, LytB should be one of the first enzymes to be transferred by nanotubes, otherwise the cell-to-cell connection cannot be made.

This is a very interesting point. We currently do not have an assay that measures the kinetics of protein transferred via nanotubes. In fact, we could see emerging nanotubes (before they are connected to the neighbouring cell) that harbour a strong signal from LytB, supporting this possibility (e.g. Reviewers' Fig 1). We included this point in the discussion (p14, Lines 7-9).

5) At the beginning of the results section, the authors mentioned that they tested the role of Lyt proteins in nanotube formation. I miss here a logic rationale that led the authors to follow up these enzymes.

We previously reported that LytB and LytC are enriched in a fraction containing partially purified nanotubes (Dubey et al., 2016). As such, the next logical step was to examine the involvement of these proteins in nanotube formation. This is mentioned in the Introduction (p4, Line 12-14), and is highlighted in the opening section of the Results (p6, Lines 4-6).

6) The signal in the vCD image in the right column of the figures sometimes does not correspond to the signal that is in the real microscopy images. In general, the localization of the signal is much clearer in the microscopy images than in the vCD image. Some readjustments are needed.

The presented vCD (low-kV, High contrast) images are unadjusted original images obtained from the XHR-SEM analysis using a vCD detector to visualize the gold particles. The vCD detector detects the backscattered electrons from the gold particles, which are much less evident in the secondary electron mode [Through lens detector- secondary electron (TLD-SE)]. At first, we acquired the image of cells and nanotubes using TLD-SE and then the same field was imaged using the vCD detector. This information was added to the Method section (Supplementary Information, p5, Lines 8-11). The images on the left (e.g. Fig. 3) are overlays of these two images. In the revised version, we provide the original TLD-SE images to clarify this point (Fig S5, S6, S7, S8, S11).

7) Fig 1, 3 and 4 needs additional data in supplemental material showing an entire EM field of the cells. this way, the reader has a better perception of how efficient was the localization of the signal in the nanotubes.

In the revised version, fully captured fields were added to address this point [Fig S1 (for Fig 1), Fig S5 (for Fig 3) and Fig S11 (for Fig 4)].

Reviewer #2 (Remarks to the Author):

In the manuscript “Donor-delivered cell wall hydrolases facilitate nanotube penetration to recipient bacteria”, Baidya et al. describe a putative mechanism of how bacterial nanotubes deliver cargo to the cytoplasm of recipient cells via controlled cell wall breaches.

If correct, this mechanism would expand our view of bacterial cell-to-cell communication and provide a mechanistic explanation for the formation of the still mysterious nanotubes. This would be interesting to a broad readership. The interplay between LytC and LytB (though somewhat preliminary) is very interesting even without the nanotube context. In essence, I think there are some very interesting threads here, but the decisive experiment (directly demonstrating migration of lytB/C along nanotubes into recipient cells as well as cell wall remodeling in recipient and donor cells) is missing. I also have some concerns about experimental data and their interpretation. Please see my detailed comments below.

1. The conclusions are often overstated. The end of the introduction, for example, states “LytB molecules were found to travel along...” and “can cooperate to promote nanotube penetration”. The conclusion of “traveling” would require time-resolved imaging of the dynamics of fluorescently-labelled LytB. Nanotube penetration is not directly demonstrated, but hypothesized from indirect data. The abstract states “the fewer nanotubes of the lytBC mutant were deficient in penetrating” – again, no evidence for nanotube penetration is presented. These conclusions should thus be weakened to indicate that they are still hypotheses/speculation.

According to the Reviewer's request, we toned down our statements regarding this issue in the: Abstract, Introduction (p4, Lines 20-22), Results (p10, Lines 5-7), and Discussion (p13, Lines 11-12).

2. Fig. S1D. The mutants' deficiency in intercellular trafficking are not clear to me. The only difference I can make out is between Δ lytBC and the rest, but these spots look like dried cell material. I am not sure I would interpret the WT as successful protein exchange. The lighter spot for the Δ lytBC mutant might as well reflect a lower cell density in the spotted mutant volume.

The assay for protein exchange was previously established in our laboratory as a faithful reporter for nanotube formation (Bhattacharya et al., 2019; Dubey and Ben-Yehuda, 2011; Dubey et al., 2016). Nevertheless, to clarify the use of the assay for molecular exchange, we provide improved experimental data showing cell viability by GFP to indicate that the spots contain living cells. As a control, the cells were spotted on the LB plates and similar densities were obtained from wt and mutant cells, indicating equal amounts of cells (Reviewers' Fig 2). Additionally, no growth defect was detected for the Δ lytBC mutant (Fig S14A).

Likewise, the plasmid exchange data are very difficult to interpret. Are the small spots really all colonies? Would they be bigger if you incubated the cell for a longer time?

The plasmid transfer assay is based on colonies appearing after selection with three different antibiotics (Bhattacharya et al., 2019; Dubey and Ben-Yehuda, 2011; Dubey et al., 2016) (legend to Fig S2). To address the Reviewer's comment, we now provide a detailed analysis describing this assay and its outcome (Fig S2).

Could they simply be abortive growth of a subpopulation due to leftover donor enzyme that degrades antibiotics in the spot (this could be tested by repurifying colonies on selective plates)?

We ruled out this possibility by re-streaking the colonies obtained after the plasmid transfer experiment on the triple antibiotic selective plate, showing acquisition of genetic information encoded by the plasmid (Dubey and Ben-Yehuda, 2011; Dubey et al., 2016). This analysis is now included in detail in Fig S2.

Also, in these assays, how can you distinguish between nanotube-mediated transfer and other means of transformation (e.g., DNA uptake from lysed cells). Did they repeat this assay in a competence(-) background?

To address the Reviewer's concern, we performed the assay with a competence deficient mutant (Δ comK). As shown previously, competence deficiency has no effect on nanotube-mediated molecular exchange (Dubey et al., 2016) (Reviewers' Fig 2). Importantly, the assay, performed in Δ CORE background as a negative control, showed no molecular exchange (Bhattacharya et al., 2019) (Reviewers' Fig 2).

3. Page 6, line 18/19. What is your explanation for the lack of GFP fluorescence? Did you try any other fluorophores/HALO tags etc.? A fluorescently-labeled LytB would be very useful to prove the main point of the paper.

LytB harbours an N-terminal signal peptide and it is a secreted protein. Secreted GFP often lacks fluorescence due to misfolding upon emerging from the translocon. We have tried several other tags to observe LytB including YFP, sfGFP and Dronpa, but no fluorescence was detected. We overcame this obstacle by using specific anti-GFP antibodies and immuno-fluorescence microscopy (Fig 2; Fig S4D; Fig S11D).

4. Nanogold labeling in general: E.g., Figure 3 – the schemata often omit background signal. While I understand why this is done, I think you should include background as well, otherwise the schema becomes an idealized version of the actual image, which means a decrease in information. For example, in Fig. 3A, many gold particles can be seen in a and b, but are not drawn in c.

In the revised version, the background signals were added to all schematics (Fig 3; Fig 4; Fig S6D).

5. I am not sure about the enrichment of gold particles near presumed nanotube attachment sites (e.g. in the *Staph aureus* experiment, Figure 3C) – could this be quantified somehow (density/random square as a function of distance from the attachment site)?

To address the Reviewer's comment, we now quantitated the gold particles on the *Sa* cell surface by slicing them into two halves: a proximal half, containing the nanotube attachment site, and a distal half, opposing the nanotube attachment site (Fig S9). The results strongly support our model.

6. Page 9, line 7 ff., “the truncation abolished the placement of the protein...” – I would not use “abolish” since both the images (Fig. 4C b) and the quantification (Fig. 4 F) do reveal a few *lytB* particles.

The text was changed from "abolished" to "largely reduced" (p9, Line 19).

7. Page 9 line 15 – I am not sure the signal on *Sa* is significantly lessened – it is hard to differentiate background from signal in Fig. 4E c. There are many gold particles in and around the *Sa* cells here, which are not drawn into the schema in d (see also my point 4 above). The authors might have determined that these are clearly background, but if so, please state in the text how this was determined. In this context, how variable is gold labeling in these assays? Please add statistics to the quantifications.

According to the Reviewer's comment, the amount of background signal from each image was quantitated by randomly selecting multiple areas lacking bacteria, which were equal in size to *Sa* cells. The average amount of gold particles was deducted from the amount of signal obtained from the *Sa* cell surface. These results are now presented in Fig 4G and the analysis is described in the Methods section (Supplementary Information p5, Lines 12-19).

8. Page 9, line 19 – I would not write “infer” here, which would imply a basis in data or direct observations. The last bit of the sentence includes the statement “...to enable efficient nanotube penetration”. Penetration is not shown, so I would use “speculate” or “hypothesize” instead of “infer”.

The text was modified from "infer" to "suggest" (p10, Lines 5-7).

Reviewer #3 (Remarks to the Author):

In this study, the authors are following up on their previous work on the intercellular membranous nanotubes that form between *Bacillus* and other species. Here, they address the question of how these nanotubes are able to cross the peptidoglycan cell wall of the producer and receiver cells. In particular, the focus is on a pair of proteins present in/on the nanotubes structures, *LytB* and *LytC*. The authors convincingly show that *LytBC* are important for nanotube formation and that *LytB* can

be transferred between cells. This is a significant finding as it sheds insight into the mechanism of this form of cell-to-cell interaction.

While the main finding of the role of LytB and LytC in tubule formation is solid, I'm very hesitant about the connection with and transfer into diverse species part of their findings.

1) - First of all, unless I'm overlooking something, the only evidence for the enzymatic activity of LytC that is shown is the need for LytB.

It was previously shown that LytC activity is enhanced by LytB (Herbold and Glaser, 1975). Consistently, we provide evidence that LytC by itself has a detectable activity that is enhanced by LytB (Fig 5; Fig 6; Fig S13).

2) Is it possible that LytBC are just "structural" proteins needed for the formation of the nanotubes?

Previous studies and our assays suggest that LytB and LytC are classical lytic proteins and not structural components of the nanotubes or other external organelles (Herbold and Glaser, 1975). Further, LytC molecules rarely localized to nanotubes (Fig S7B-S7C). Similar proteins were shown to be important in the extrusion of additional organelles, but not to be a structural part of these organelles (Arends et al., 2013; Baron et al., 1997; Bobrovskyy et al., 2018; DeWitt and Grossman, 2014; Garcia-Gomez et al., 2011; Hirano et al., 2001). Furthermore, we constructed LytC "catalytically dead" mutants and found them to be deficient in nanotube formation, thus substantiating this assumption (see comment 4 below).

3) While there is precedent for cell wall remodeling to be needed for production of secretion systems/surface structures, is it definitely LytC that is doing it?

We assume that LytB and LytC are involved in the formation of nanotubes, as a *lytBC* mutant is largely deficient in nanotube formation and molecular exchange (Fig 1; Fig S1; Fig S2), and both proteins were associated with the nanotube biochemical fraction (Dubey et al., 2016). In addition, LytB was found to localize at sites of nanotube attachment (Fig 3), substantiating this point. Finally, LytC deficient in amidase activity is deficient in nanotube formation (see comment 4 below).

4) The authors do show that LytC can cleave cell walls, but does it *need* to in order to form the nanotubes? The authors don't really show this. It would be nice to see that a catalytically dead LytC has the same phenotype as a LytC deletion.

To address the Reviewer's comment, we constructed multiple strains lacking the LytC amidase activity by site-directed mutagenesis. In all the constructed strains, the continual nanotube phenotype was observed, and the intercellular molecular exchange was significantly reduced, similarly to Δ *lytC*. These results are now included in the revised version (Fig S3; p6, Lines 16-18).

5) I'm not really convinced that the nanotubes formed between *Bacillus* and *Vibrio* are actually connecting with *Vibrio* as the text seems to imply. LytB is not being transferred, so how do you know that the connection is actually being made? The picture in fig S4 looks more like membrane fusion (or maybe EM artifacts) than the clearly defined nanotubes that form between the gram-positive organisms.

To address this Reviewer's concern, we now provide improved images to substantiate the notion that nanotubes are formed between *Vc* cells (Fig S8D). *Vc* is a Gram negative bacterium, and as such,

does not have a surface-exposed cell wall and was therefore used as a negative control. We could not detect molecules of *Bs* LytB on the *Vc* surface, whereas they were plentiful on the surface of *Bm* and *Sa* (Fig 3).

6) At the very least, I would like to see a wt vs Δ lytBC mutant with *Vc* to confirm that the connections to *Vc* like shown in fig S4 go away. Alternatively, maybe the authors could show that something is actually being transferred to the *Vc* in the context of these lytBC experiments.

To address the Reviewer's comment we now provide a detailed and quantitative analysis of nanotube formation between *Bs* and *Vc* (Fig S10). Unfortunately, we do not yet have an assay that reports molecular exchange between these two species.

7) Continuing with that point, I would like to see confirmation whether LytC in the receiver cell is or is not needed for nanotube connection to be made. i.e. KO the LytC from *Sa* and see if you still get nanotubes + LytB transfer. Or a use a *Bs* that is Δ ymdB Δ lytBC that doesn't produce nanotubes, can it be a receiver still.

We show that LytC in the receiver cell helps in nanotube formation between *Bs* and *Bm* cells. Accordingly, when both species lacked LytC, the amount of nanotube formation between *Bs* and *Bm*, as well as the nanotube-mediated killing of *Bm* was significantly reduced (Fig 7; Fig S14).

8) Related to the previous point, is it possible that the "continual" nanotubes seen in Fig 1, are an indicator of the Δ lytBC cells being unable to act as receiver cells, as much as it is about producing fewer nanotubes? Since the nanotubes are only visible by EM, and their dynamics aren't really visible in real time, is it possible that the reason you see a lot of nanotubes forming in the wt is that they successfully attach to receiver cells and are therefore stabilized. Meanwhile in the Δ lytBC, the nanotubes are less stable so fewer of them are visible.

This is a good point. However, this possibility seems inconsistent with the interspecies interactions where *Bs* serves as a donor and *Bm* as a recipient. In these experiments, nanotube-mediated killing of *Bm* was largely reduced by *Bs* Δ lytBC mutant in comparison to *Bs* wt, whereas in both cases the recipient was wt *Bm* (Fig 7A).

References

- Antelmann, H., Yamamoto, H., Sekiguchi, J., and Hecker, M. (2002). Stabilization of cell wall proteins in *Bacillus subtilis*: a proteomic approach. *Proteomics* 2, 591-602.
- Arends, K., Celik, E.K., Probst, I., Goessweiner-Mohr, N., Fercher, C., Grumet, L., Soellue, C., Abajy, M.Y., Sakinc, T., Broszat, M., *et al.* (2013). TraG encoded by the pIP501 type IV secretion system is a two-domain peptidoglycan-degrading enzyme essential for conjugative transfer. *J Bacteriol* 195, 4436-4444.
- Baron, C., Llosa, M., Zhou, S., and Zambryski, P.C. (1997). VirB1, a component of the T-complex transfer machinery of *Agrobacterium tumefaciens*, is processed to a C-terminal secreted product, VirB1. *J Bacteriol* 179, 1203-1210.
- Bhattacharya, S., Baidya, A.K., Pal, R.R., Mamou, G., Gatt, Y.E., Margalit, H., Rosenshine, I., and Ben-Yehuda, S. (2019). A ubiquitous platform for bacterial nanotube biogenesis. *Cell reports* 27, 334-342 e310.
- Bobrovskyy, M., Willing, S.E., Schneewind, O., and Missiakas, D. (2018). EssH peptidoglycan hydrolase enables *Staphylococcus aureus* Type VII secretion across the bacterial cell wall envelope. *J Bacteriol* 200.
- DeWitt, T., and Grossman, A.D. (2014). The bifunctional cell wall hydrolase CwIT is needed for conjugation of the integrative and conjugative element ICEBs1 in *Bacillus subtilis* and *B. anthracis*. *J Bacteriol* 196, 1588-1596.
- Dubey, G.P., and Ben-Yehuda, S. (2011). Intercellular nanotubes mediate bacterial communication. *Cell* 144, 590-600.
- Dubey, G.P., Malli Mohan, G.B., Dubrovsky, A., Amen, T., Tsipshtein, S., Rouvinski, A., Rosenberg, A., Kaganovich, D., Sherman, E., Medalia, O., *et al.* (2016). Architecture and characteristics of bacterial nanotubes. *Developmental cell* 36, 453-461.
- Garcia-Gomez, E., Espinosa, N., de la Mora, J., Dreyfus, G., and Gonzalez-Pedrajo, B. (2011). The muramidase EtgA from enteropathogenic *Escherichia coli* is required for efficient type III secretion. *Microbiology* 157, 1145-1160.
- Herbold, D.R., and Glaser, L. (1975). *Bacillus subtilis* N-acetylmuramic acid L-alanine amidase. *The Journal of biological chemistry* 250, 1676-1682.
- Hirano, T., Minamino, T., and Macnab, R.M. (2001). The role in flagellar rod assembly of the N-terminal domain of Salmonella FlgJ, a flagellum-specific muramidase. *Journal of molecular biology* 312, 359-369.

Reviewer's Fig 1: Evidence that LytB is one of the earliest nanotube-delivered proteins. *Bs* cells expressing LytB-GFP (AB144: *lytB-gfp*, $\Delta ymdB$, *amyE::P_{hyper-spank}-ymdB*, Δhag) were spotted onto EM grids and subjected to immuno-gold XHR-SEM analysis, using primary antibodies against GFP and secondary gold-conjugated antibodies. Samples were not coated before observation. Shown is an XHR-SEM image acquired using TLD-SE (through lens detector-secondary electron) for nanotube visualization (**A**), and the corresponding vCD (low-kV high-contrast detector) for gold particle (white dots) detection (**B**). Arrow indicates the emerging nanotube.

Reviewer's Fig 2: Assessing the effect of competence deficiency on molecular exchange. Pairs of a donor (SB463: *amyE::P_{hyper-spank}-cat-spec*) (CmR, SpecR) and a recipient (SB513: *amyE::P_{hyper-spank}-gfp-kan*) (KanR) parental strains (wt) were used. The investigated mutants indicated in the figure harbour the corresponding genotypes and carry the indicated null mutation in both donor and recipient strains. Donor and recipient strains were mixed in 1:1 ratio (at two concentrations x1, x0.1) and incubated in LB supplemented with 1 mM IPTG for 4 h at 37°C with gentle shaking. Equal numbers of cells were then spotted onto LB agar (Control) and LB agar containing chloramphenicol (Cm) and kanamycin (Kan). Plates were imaged after 18 hrs of incubation by colorimetric imaging and by a fluorescent reader to monitor GFP expression, as a measure for cell viability.

REVIEWERS' COMMENTS:

Reviewer #1 (Remarks to the Author):

All my comments have been addressed.

Just one minor comment: In this new version of the manuscript, the graphs contains the error bars and show quite a significant variation. I understand that this is microscopy analyses so it is subjected to certain variation but I wonder if the authors can add an explanation in the methods section of the manuscript about where such a variability comes from (number of biological replicates vs. technical replicates, sampling volume...). It would be helpful to anyone that wants to reproduce the experiments, to let them aware that certain variability should be expected and what are the possible reasons for such variability.

Reviewer #2 (Remarks to the Author):

The manuscript is much improved and all my concerns have been adequately addressed. I now endorse publication without further revisions.

Reviewer #3 (Remarks to the Author):

The authors have sufficiently addressed my concerns with the manuscript

Response to Reviewers' comments

Reviewer #1 (Remarks to the Author):

All my comments have been addressed.

Thank you.

Just one minor comment: In this new version of the manuscript, the graphs contains the error bars and show quite a significant variation. I understand that this is microscopy analyses so it is subjected to certain variation but I wonder if the authors can add an explanation in the methods section of the manuscript about where such a variability comes from (number of biological replicates vs. technical replicates, sampling volume...). It would be helpful to anyone that wants to reproduce the experiments, to let them aware that certain variability should be expected and what are the possible reasons for such variability.

All the graphs now include the actual data points according to the editor's request. This should address the reviewer's concern.

Reviewer #2 (Remarks to the Author):

The manuscript is much improved and all my concerns have been adequately addressed. I now endorse publication without further revisions.

Thank you.

Reviewer #3 (Remarks to the Author):

The authors have sufficiently addressed my concerns with the manuscript

Thank you.